# Persistence of the Immune Response to an Intramuscular Bivalent (GI.1/GII.4) Norovirus Vaccine in Adults

**DOI:** 10.3390/vaccines13010082

**Published:** 2025-01-17

**Authors:** Geert Leroux-Roels, Robert L. Atmar, Jakob P. Cramer, Ian Escudero, Astrid Borkowski

**Affiliations:** 1Center for Vaccinology (CEVAC), Ghent University and University Hospital, 9000 Ghent, Belgium; geert.lerouxroels@ugent.be; 2Departments of Medicine and Molecular Virology and Microbiology, Baylor College of Medicine, Houston, TX 77030, USA; ratmar@bcm.edu; 3Clinical Development, Takeda Pharmaceuticals International AG, Farman Strasse 11, Opfikon, 8152 Zurich, Switzerland; jakob.cramer@cepi.net (J.P.C.); ian.escudero@takeda.com (I.E.); 4Clinical Development, Coalition for Epidemic Preparedness Innovations (CEPI), Gibbs Building, 215 Euston Road, London NW1 2BE, UK; 5Clinical Development, Takeda Vaccines, 75 Sidney St., Cambridge, MA 02139, USA; 6HilleVax AG, Boulevard Lilienthal 42, Opfikon, 8152 Zurich, Switzerland

**Keywords:** norovirus, vaccine, stomach flu, gastroenteritis, virus-like particle, immunogenicity, persistence

## Abstract

Background: Major global economic and health burdens due to norovirus gastroenteritis could be addressed by an effective vaccine. Methods: In this study, 428 adult recipients of various compositions of the norovirus vaccine candidate, HIL-214, were followed for 5 years, to assess immune responses to its virus-like particle antigens, GI.1 and GII.4c. Serum antibodies and peripheral-blood antibody-secreting cells (ASCs) were measured. This report focuses on the single-dose 15/50 (µg GI.1/GII.4c) composition, which had been selected for further clinical development. Results: For single-dose 15/50 recipients (N = 105), GI.1-specific and GII.4c-specific histoblood-group antigen-blocking (HBGA) antibodies appeared to have persisted to 5 years, waning from a peak at 4 to 8 weeks, and plateauing above baseline after 3 years. From 3 to 5 years, GI.1-specific GMTs ranged between 53 (95%CI, 40–71) and 60 (95%CI, 46–77; N = 69–97) and were approximately 2-fold above the baseline GMT (24 (95%CI, 20–28); N = 105). GII.4c-specific GMTs ranged between 103 (95%CI, 77–138) and 114 (95%CI, 86–152; N = 70–97) and were above baseline, but by less than 2-fold (70 (95%CI, 53–92); N = 105). Similar kinetics were observed for pan-Ig titers and ASCs in a subset. Similar kinetics were also observed for HBGA and pan-Ig titers in recipients of other 15/50 dosages. Conclusions: Immune responses to HIL-214 in adults appear to persist for five years.

## 1. Introduction

Acute gastroenteritis due to norovirus infection represents in a significant global health burden, with high levels of morbidity across all ages, and the possibility of mortality in at-risk groups, such as the very young, very old, and the immunocompromised [1]. Prophylactic vaccination is a potential strategy for preventing norovirus gastroenteritis, and several candidate vaccines are in clinical development [2]. The most advanced candidate is HIL-214 (formerly TAK-214), a bivalent virus-like particle (VLP) vaccine containing two VLPs targeting the predominant norovirus genogroups responsible for most human disease, GI.1 and GII.4 [3]. As antigenic drift has resulted in multiple variants of GII.4, the vaccine GII.4 VLP consists of a consensus sequence (GII.4c) constructed from three different naturally occurring GII.4 genotype variants, 2002 (Houston), 2006a (Yerseke), and 2006b (Den Haag). The construct is intended to induce a broad immunity against diverse variants within the GII.4 genotype [4,5].

Several clinical trials have demonstrated immunogenicity and an acceptable safety profile in adults and older-adult volunteers for various compositions of HIL-214, which contain different ratios of the two VLP antigens and are adjuvanted with aluminum hydroxide (Al(OH)_3_) with or without monophosphoryl lipid A (MPL) [6,7,8]. From those trials, a single-dose composition, comprising 15 μg GI.1 and 50 μg GII.4c VLPs with 0.5 mg Al(OH)_3_, was selected for adults. No added benefit to immunogenicity was observed from a second dose or from the inclusion of MPL in the vaccine composition [7,8]. This single-dose composition was subsequently evaluated in a proof-of-concept study involving US Navy recruits, where the vaccine efficacy against moderate-to-severe gastroenteritis associated with any norovirus genotype was estimated to be 61.8% (95.01% CI, 20.8–81.6) [9].

The current Phase 2 study was designed to assess the long-term immunogenicity of HIL-214 in adult participants pooled from three previous clinical trials, up to five years after vaccination. This Phase 2 study considered fifteen different regimens because two of the three trials were dose-finding in design [7,8]. However, in this report, the immunogenicity analysis focuses on the 15/50/0 composition (i.e., dose quantities in µg of GI.1 VLP/GII.4c VLP/MPL) administered as a single dose, because these VLP quantities and this regimen were selected for further clinical development. The analyses of the one-dose 15/50/15 regimen and the two-dose 15/50/0 regimen were conducted to support the analysis of the one-dose 15/50/0 regimen, and as with the one-dose 15/50/0 regimen, both regimens were used in at least two of the previous trials [6,7,8]. The primary analysis relied on an assay for titering serum antibodies that block VLP binding to histoblood-group antigens (HBGA)—sugar moieties on intestinal enterocytes that can bind human norovirus [10]. Although human norovirus can now be cultured in intestinal organ cultures, the HBGA-blocking assay is considered as a quasi-neutralization assay that remains better suited for high-throughput processing [11].

## 2. Methods

### 2.1. Study Design

This long-term follow-up Phase 2 study, designated as NOR-213, is registered on ClinicalTrials.gov (NCT03039790) and EudraCT (2016-004288-37). The study population consisted of adults who had previously participated in 1 of 3 trials (NOR-107, NOR-204, and NOR-210) conducted at 11 trial sites—2 in Belgium and 9 in the United States—between 21 February 2017 and 22 July 2021. The vaccine candidate in those trials, initially developed and supplied by Takeda Vaccines as TAK-214, is currently being advanced through clinical development by HilleVax as HIL-214. All participants completed and signed a new informed consent for this study. The protocol was approved by the ethics committees of the respective study sites and conducted according to current guidelines of the International Council for Harmonisation of Technical Requirements for Pharmaceuticals for Human Use and good clinical practice. The three initial trials were as follows:

NOR-107 (NCT02038907 [7]) was conducted at 2 sites in Belgium between 28 March 2014 and 19 June 2015 in 420 adults aged 18 to 64 years. Participants received 1 or 2 doses of 11 different compositions of GI.1 and GII.4c with or without 15 µg or 50 µg of MPL and/or 167 µg or 500 µg Al(OH)_3_ as adjuvants—see Table 1. Participants from NOR-107 entered the present study during their third year after vaccination.

NOR-204 (NCT02661490 [8]) was conducted at 10 sites in the US, mainly involving 294 older adults aged ≥60 years, with a control group of 26 adults aged 18 to 49 years, who received 1 or 2 doses of 15 µg GI.1 and 50 µg GII.4c with 500 µg Al(OH)_3_, with or without 15 µg MPL—see Table 1. Participants from NOR-204 entered the present study during their second year after vaccination.

NOR-210 (NCT02475278 [6]) was conducted at 1 site in the US in 50 adults aged 18 to 49 years, who received 1 dose of 15 µg GI.1 and 50 µg GII.4c with 500 µg Al(OH)_3_. Participants from NOR-210 entered the present study during their second year after vaccination.

For inclusion in the present study, male or female volunteers aged over 18 years must have previously received at least 1 intramuscular dose of HIL-214 in 1 of the previously mentioned trials and must not have received any investigational product from any other clinical trial within 30 days prior to the start of this study. Acceptance into the study was also contingent on the investigator’s assessment that the volunteer would complete the follow up, provide the required blood samples, and had no behavioral or cognitive impairment or psychiatric condition that might interfere with their participation. For inclusion into the initial trials, all participants had been required to be in good health, or in a medically stable health condition.

### 2.2. Objectives

Primary and secondary objectives were to evaluate the humoral immune response annually for up to five years after at least one dose of HIL-214, as measured by HBGA-blocking and pan-Ig assays, respectively. An assessment of antibody-secreting cell (ASC) frequency was included as an exploratory objective.

### 2.3. Endpoints

The primary immunogenicity endpoints were the geometric mean titers (GMTs) of GI.1-specific and GII.4c-specific HBGA-blocking antibodies. Secondary endpoints were GMTs of GI.1-specific and GII.4c-specific pan-Ig antibodies. Exploratory endpoints included the frequency of GI.1-specific and GII.4c-specific ASCs. All primary, secondary, and exploratory evaluations were performed at yearly intervals up to 5 years after primary vaccination.

### 2.4. Procedures

At the initial trial visit, a baseline blood sample was taken from each participant, with further blood draws occurring on Day 28 or 56 (one month after the first or second dose) and at six months (Day 208 or 211 depending on the study). In this extension study, blood samples were drawn at two years and then at one-year intervals until the fifth anniversary of their first HIL-214 dose. At each study visit, participants were interviewed to determine if any serious adverse events (SAEs) had occurred since the previous visit. These SAEs were assessed by the investigator for severity and potential causality related to the vaccine. Participants were instructed to report any SAE immediately or within 24 h to the investigator for further investigation. A data monitoring committee, which was already in place to oversee the risks, benefits, and scientific validity of the entire HIL-214 development program, had oversight of the present study.

### 2.5. Immunogenicity Assays

Sera were prepared immediately from blood draws at each study visit and shipped at −20 °C to the PPD Bioanalytical Laboratory (Richmond, VA, USA). HBGA-blocking antibodies were measured as described previously [7,10]. Pan-Ig antibodies were measured by the enzyme-linked immunosorbent assay. Both assays were performed with 2-fold dilutions of serum samples, where a titer represents the reciprocal of the dilution. ASCs were identified in an ELISpot assay of cultured peripheral-blood mononuclear cells (PBMCs), largely based on a previously published method [12] and performed as a service by CEVAC—Ghent University (Ghent, Belgium). Briefly, memory B cells were induced to differentiate into plasma cells by culturing PBMCs with R848 (InvivoGen, Toulouse, France) and recombinant human IL-2 (R&D Systems, Abingdon, UK) for 5 days at 37 °C/5%CO_2_. Cells were then transferred into (96-plate) culture wells coated with 100 µL of GI.1 or GII.4c VLPs at 10 µg/mL or with 100 µL goat–anti-human IgG at 50 µg/mL (Affinipure, Jackson ImmunoResearch Europe Ltd. Ely, UK) to enumerate GI.1- or GII.4c-specific antibodies or IgG-secreting plasma cells, respectively. The antibody/antigen spots formed were detected by a conventional immunoenzymatic procedure. The results were expressed as frequencies of GI.1- and GII.4c-specific memory B cells per million of IgG-producing memory B cells.

### 2.6. Statistics

Data from participants who received a given HIL-214 composition were pooled for each time point for the analysis (see Table 1). For this trial, there were two analysis sets defined: the full analysis set (FAS), which consisted of all participants with data from at least one follow-up time point, and the per-protocol set (PPS), which included all participants in the FAS with no major protocol deviations that could potentially confound the primary (immunogenicity) endpoint. An additional post hoc analysis of the immunogenicity data was performed, excluding data from participants considered to have had a natural norovirus infection from the time point after the infection occurred. Natural infection was indicated by a ≥4-fold increase in titer, over the most recent previously collected titer, for either antibody assay (HBGA-blocking or pan-Ig) or either GI.1 or GII.4c specificity. A third subset, the ASC subset, consisted of participants originally included in the cell-mediated immunity (CMI) subsets from the NOR-107 and NOR-204 trials and was analyzed descriptively for ASC frequencies. Safety data, including the occurrence of serious adverse events or deaths, are presented descriptively. All statistical analyses were generated using SAS Version 9.4 (SAS Institute Inc., Cary, NC, USA).

## 3. Results

### 3.1. Disposition and Demographics

In total, 528 individuals were enrolled into this long-term follow-up study comprising participants from NOR-107, NOR-210, or NOR-204 clinical trials (Figure 1; Table 1). Fifty-three participants discontinued the study, including twenty-two lost to follow-up, twenty-one who withdrew, six who died, and four who discontinued for unspecified reasons. In both the FAS and the PPS, 105 participants were 1-dose 15/50/0 recipients (Arm 8A), 67 were 2-dose 15/50/0 recipients (Arm 8B), and 52 were 1-dose 15/50/15 recipients (Arm 5A). A subset analyzed for ASCs consisted of 234 participants exclusively from NOR-107 and NOR-204, including 34 recipients from Arm 8A who received a single dose of 15/50/0.

Overall, the mean age was 54 years, with the majority of participants (77%) aged between 18 and 64 years (Appendix A). Sixty-one percent of participants were female. In terms of race, most participants identified as White (97%). In terms of ethnicity, most participants (65%) did not report a category, and 32% reported being not Hispanic or Latino. For recipients of the 15/50 GI.1/GII.4c compositions, mean ages differed: in Arm 8A, the mean age was 54 years, compared with 64 and 62 years in Arms 8B and 5A, respectively (Table 2). Additionally, fewer participants were female in Arm 8A compared with Arms 8B and 5A (48% versus 57% and 65%, respectively). During the study, 421 out of 528 participants (80%) reported ongoing medical conditions. The four most frequently reported conditions, categorized by the MedDRA System Organ Class, were (1) musculoskeletal and connective tissue disorders (35%), (2) vascular disorders (30%), (3) immune system disorders (30%), and (4) metabolism and nutrition disorders (27%). Additionally, fifteen pregnancies were reported during the study.

### 3.2. Immunogenicity by HBGA-Blocking Antibodies (Primary Objective)

In the one-dose 15/50/0 recipients (Arm 8A), GI.1-specific and GII.4c-specific HBGA-blocking antibodies appeared to persist up to five years post-vaccination. These antibodies increased from baseline to peak at four to eight weeks, then plateaued above baseline after three years (Figure 2), and showed greater fold changes compared with baseline for GI.1 GMTs than for GII.4c GMTs. This latter difference stemmed from lower baseline GI.1 GMTs, likely reflecting less recent immune exposure and lower prevalence of GI.1 genotypes compared with GII.4 genotypes. Hence, from 3 to 5 years, GI.1 GMTs ranged between 53 (95% confidence interval (95%CI), 40–71) and 60 (95%CI, 46–77; N = 69–97), remaining below the peak GMT of 229 (95%CI, 161–326; N = 81), but approximately 2-fold higher than baseline GMTs of 24 (95%CI, 20–28; N = 105). GII.4c GMTs ranged between 103 (95%CI, 77–138) and 114 (95%CI, 86–152; N = 70–97), below the peak GMT of 623 (95%CI, 372–1042; N = 53), yet consistently above baseline, but by less than 2-fold (70 (95%CI, 53–92); N = 105). After 3 to 5 years, median GI.1-specific titers (122–136) and GII.4c-specific titers (52–70) were also higher compared with their respective baseline median titers of 59 and 15.

The HBGA-blocking GMTs at three to five years were also higher than baseline when excluding data from vaccine recipients considered to have had a natural norovirus infection between two and five years after vaccination (i.e., ≤19% of one-dose 15/50/0 recipients had either a GI.1-like or GII.4-like infection; Appendix A). For recipients of the one-dose 15/50/0 regimen (Arm 8A), the trends in the HBGA-blocking GMTs remained consistent when analyzed by initial trial (i.e., NOR-107, NOR-210, and NOR 204) and by age within NOR-204 (Appendix A). All GI.1-specific HBGA-blocking GMTs at 3 to 5 years were higher than those at baseline, except for 1 GMT (in the 60- to 94-year age group from NOR-204), and GII.4c-specific HBGA-blocking GMTs at 3 to 5 years were also higher than those at baseline.

The trends in the HBGA-blocking GMTs of recipients of the two-dose 15/50/0 (Arm 8B) and the one-dose 15/50/15 (Arm 5A) regimen were similar to those of the one-dose 15/50/0 recipients (Figure 2). In recipients of the two-dose 15/50/0 regimen, GI.1-specific and GII.4c-specific HBGA-blocking GMTs measured three to five years after dosing were higher than at baseline. For GI.1, GMTs ranged between 69 (95%CI, 51–95) and 82 (95%CI, 60–110; N = 40–62) after 3 to 5 years, compared with 25 (95%CI, 20–32; N = 67) at baseline. For GII.4c, GMTs ranged between 117 (95%CI, 78–174) and 157 (95%CI, 110–223; N = 41–62) after 3 to 5 years, versus 91 (95%CI, 66–127; N = 67) at baseline. In recipients of the 1-dose 15/50/15 regimen (Arm 5A), GI.1 GMTs 3 to 5 years after dosing were higher than at baseline (41 (95%CI, 28–61) to 50 (95%CI, 37–69; N = 31–47) versus 20 (N = 52)). GII.4c GMTs were higher than baseline only at 3 years within the 3-to-5-year period (119 (95%CI, 80–176; N = 31–47) versus 92 (95%CI, 61–141; N = 52)).

### 3.3. Immunogenicity by Pan-Ig Antibody-Binding Assay (Secondary Objective)

GI.1- and GII.4c-specific pan-Ig-binding antibodies also appeared to persist up to five years after vaccination, similar to what was observed with HBGA-blocking antibodies (Figure 3). In recipients of the one-dose 15/50/0 regimen (Arm 8A), the magnitude of the fold change after three to five years compared with baseline was greater than two-fold for both GI.1 and GII.4c GMTs. For GI.1, GMTs after 3 to 5 years were 1.7 × 10^3^ (N = 71–97), whereas the GMT at baseline was 0.5 × 10^3^ (N = 105). For GII.4c, GMTs after 3 to 5 years ranged between 1.4 and 1.7 × 10^3^ (N = 71–97), whereas the GMT at baseline was 0.6 × 10^3^ (N = 105).

For recipients of the one-dose 15/50/0 regimen (Arm 8A), the trends in the pan-Ig GMTs remained consistent when analyzed by the initial trial and by age in NOR-204. All GI.1 GMTs at 3 to 5 years were higher than those at baseline, and except for 1 GMT (in the 18- to 49-year group from NOR 204), all 3- to 5-year GII.4c GMTs were higher than those at baseline (Appendix A).

The trends in the pan-Ig GMTs were similar when excluding data from vaccine recipients who were considered to have had a natural norovirus infection between two and five years after vaccination (Appendix A).

The trends in the pan-Ig GMTs of recipients of the two-dose 15/50/0 regimen (Arm 8B) and the one-dose 15/50/15 regimen (Arm 5A) were similar to those of the one-dose 15/50/0 recipients (Figure 3). In recipients of the two-dose 15/50/0 regimen (Arm 8B) and the one-dose 15/50/15 regimen (Arm 5A), all GI.1 GMTs were approximately two-fold higher, and all GII.4 GMTs were less than two-fold higher, respectively, three to five years after dosing, compared with the baseline.

### 3.4. Antibody-Secreting Cell Responses (Exploratory Objective)

There was evidence of B-cell memory to HIL-214 vaccination three to five years after dosing (Figure 4). For the 1-dose 15/50/0 recipients (Arm 8A) in the ASC subset, stratified by former trial, all 3- to 5-year GI.1-specific median ASC frequencies, except 1 median (from the 60- to 94-year group from NOR-204), were greater than those at baseline. Similarly, all 3- to 5-year GII.4c-specific ASC frequencies, except 2 medians (from NOR-107 and the 60- to 94-year group from NOR-204), were greater than those at baseline. From 3 to 5 years, GI.1 median ASC frequencies ranged between 0.6 and 2.9 × 10^3^ per million ASCs (N = 5–16), compared with baseline median frequencies ranging between 0.4 and 0.8 × 10^3^ per million ASCs (N = 5–17). GII.4c median ASC frequencies ranged between 0.4 and 1.0 × 10^3^ per million ASCs (N = 5–16), whereas baseline median frequencies ranged between 0.1 and 0.6 × 10^3^ per million ASCs (N = 5–17).

### 3.5. Safety

Although there was no safety objective in the study, life-threatening and fatal SAEs, potential vaccine-related SAEs, potential immune-mediated events, events of medical significance, and any procedure-related medical occurrences were monitored during the follow-up period (1–5 years after vaccination; Table 3). Overall, there were 26 adverse events reported by 18 participants (3.4%), including 21 SAEs reported by 16 participants, including 6 deaths. One SAE (metastatic renal cell carcinoma) led to the withdrawal of one participant, and it was ultimately fatal. The investigators did not consider that any of the AEs, including SAEs and deaths, were related to the study vaccinations administered in the previous trials. However, one adverse event was causally related to the blood draw in this trial, and this was reported as a mild-intensity post-procedural complication.

## 4. Discussion

This two- to five-year follow-up Phase 2 study of adult HIL-214 recipients from three previous clinical trials (NOR-107, NOR-210, and NOR-204) identified evidence that norovirus vaccine immunogenicity persisted up to five years, notably marked by HBGA-blocking titers, pan-Ig titers, and ASC frequencies remaining above pre-vaccination baseline values. The analysis was descriptive and focused on the 1-dose 15/50/0 arm, which comprised 105 of the 528 participants recruited. This dosage was selected as the preferred one for adults in two previous trials (NOR-107 and NOR-204). Participants, enrolled at two sites in Belgium and nine sites in the US, were considered immune experienced to norovirus. The mean age at vaccination was 54 years, with 77% of participants aged between 18 and 64 years, 97% identifying as White, and 61% being female. SAEs reported during follow-up were considered unrelated to vaccination and consistent with expectations based on the demographics and prior medical conditions of the study population.

The evidence supporting persistent vaccine immunogenicity was derived from primary and secondary analyses of serum HBGA-blocking titers and pan-Ig titers, respectively. Although no inferential statistics were applied and the GMTs from three to five years post-vaccination were only approximately two-fold or less than two-fold greater than baseline (pre-vaccination), several aspects of the analyses reinforced the study’s main conclusion. These aspects included (i) the GMT relationship identified at the three-, four-, and five-year time points, suggesting that GMTs had plateaued at a steady-state level higher than the pre-vaccination baseline, (ii) the GMT relationship generally held when analyzed with respect to subgroups based on previous trial allocation, (iii) the GMT relationship persisted when data from putatively infected participants were excluded, and (iv) similar GMT relationships were observed with related dosages (two-dose 15/50/0 and one-dose 15/50/15). Additionally, the analysis of a smaller ASC subset suggested that circulating B cells in peripheral blood may have contributed to the sustained production of serum antibodies throughout five years.

The absence of a placebo control group for monitoring long-term changes from baseline titers due to natural infections during the study was a limitation of the analysis. Estimates suggest a norovirus disease rate of approximately 5% of the adult population per year [13,14,15]. The baseline GMTs prior to vaccination reflected immune experience to norovirus, particularly to GII.4-specific over GI.1-specific genotype norovirus strains. The mitigation step to remove data by the final year from up to 19% of vaccine recipients with putative natural GI.1-specific or GII.4-specific norovirus infections inevitably resulted in lower post-vaccination GMTs, as a putative infection was defined by an increase in a GI.1- or GII.4c-specific titer. This mitigation may have biased the results toward underestimating the true difference from baseline, especially with GII.4c-specific GMTs. Nevertheless, as stated above, the adjusted GMTs generally remained above the respective pre-vaccination baselines.

Another limitation was that antibody titers and ASC frequencies (in a smaller subset) in peripheral blood provided only a narrow view of anti-norovirus immunity and an indirect assessment of cellular immune memory [16]. Nevertheless, it is reasonable to assume that both factors would play a role in an anamnestic response to norovirus infection. Moreover, HBGA-blocking antibodies (the primary endpoint measure) may be clinically relevant because these antibodies potentially neutralize norovirus infection of enterocytes in the gut, and the incidence of norovirus gastroenteritis is lower in individuals with mutations that prevent the synthesis of HBGA [11,17,18,19,20]. However, a clear association with responses to vaccination and the prevention of norovirus gastroenteritis remains to be identified [9,21].

## 5. Conclusions

This study provided evidence that in adults, the immunogenicity of a single dose of the norovirus vaccine HIL-214 persisted up to five years. This was indicated by HBGA-blocking titers, pan-Ig titers, and ASC frequencies remaining above pre-vaccination baseline values.

## Figures and Tables

**Figure 1 vaccines-13-00082-f001:**
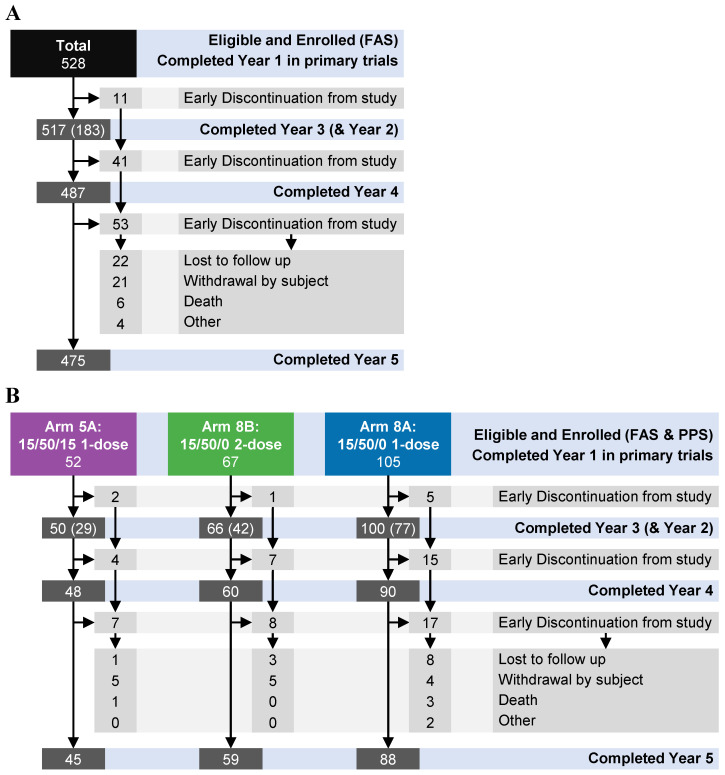
Schematic description of participant disposition for all eligible and enrolled participants entered in the follow-up study from the initial trials NOR-107, NOR-210, and NOR-204. Participants from NOR-210 and NOR-204 entered the follow-up study at the beginning of Year 2 (i.e., one year after vaccination), whereas participants from NOR-207 entered at the beginning of Year 3. The disposition is shown for (**A**) the full analysis set (FAS) for the entire study population and (**B**) for the FAS for Arms 5A, 8B, and 8A. The light-grey boxes indicate participants who discontinued, and the reasons for discontinuation. The numbers in the light-grey boxes represent cumulative totals, in accordance with the directions of the arrows. PPS = per-protocol set.

**Figure 2 vaccines-13-00082-f002:**
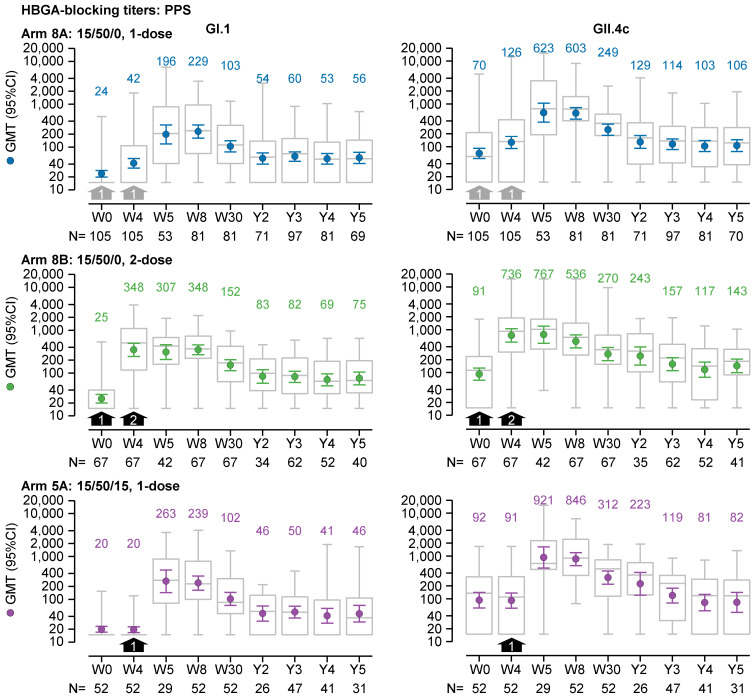
GI.1-specific and GII.4c-specific HBGA-blocking geometric mean titers (GMTs; left and right graphs, respectively) over a five-year period from prior to dosing (Week 0, W0) in the preceding trials to the end of the present follow-up study (Year 5, Y5) in the per-protocol set (PPS) without data from participants putatively infected with norovirus (with either a GI.1-like or GII.4-like genotype and identified by a ≥4-fold increase in antibody titer at a subsequent visit during the follow-up period). Three dosage regimens are shown: (i) one dose of 15/50/0 (either at W0 or W4; grey arrows), (ii) two doses of 15/50 (at W0 and W4; black arrows), and (iii) one dose of 15/50/15 at W4 (black arrows). GMTs and 95% confidence intervals are represented by colored round symbols (and value above the plot) and error bars, respectively. Medians, 25 and 75 percentiles, and minimum and maximum values are represented by grey box and whisker plots.

**Figure 3 vaccines-13-00082-f003:**
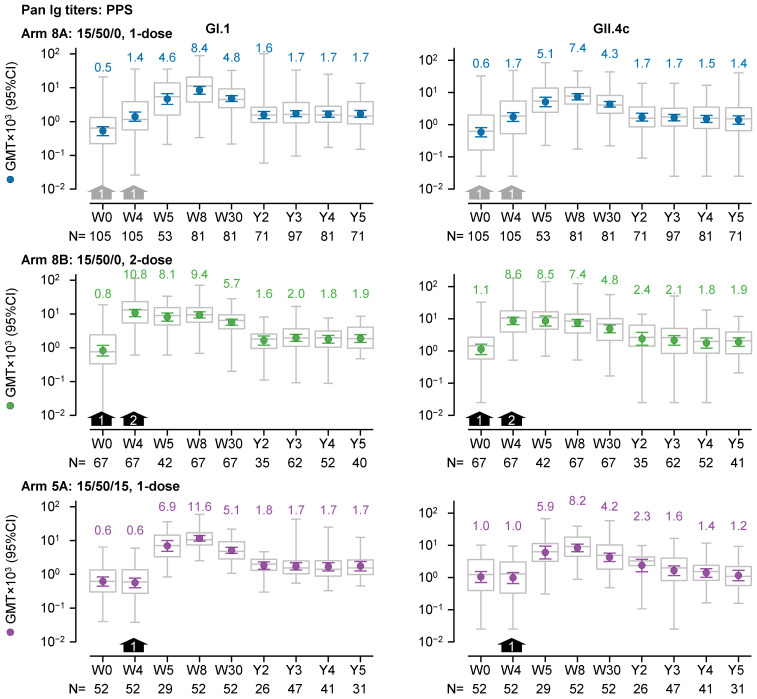
GI.1-specific and GII.4c-specific pan-Ig geometric mean titers (GMTs; left and right graphs, respectively) over a five-year period from prior to dosing (Week 0, W0) in the preceding trials to the end of the present follow-up study (Year 5, Y5) in the per-protocol set (PPS). Three dosage regimens are shown: (i) one dose of 15/50/0 (either at W0 or W4; grey arrows), (ii) two doses of 15/50/0 (at W0 and W4; black arrows), and (iii) one dose of 15/50/15 at W4 (black arrows). GMTs and 95% confidence intervals are represented by colored round symbols (and value above the plot) and error bars, respectively. Medians, 25 and 75 percentiles, and minimum and maximum values are represented by grey box and whisker plots. Note, the GMTs in the figure are expressed in a scientific notation that includes a 10^3^ multiplier.

**Figure 4 vaccines-13-00082-f004:**
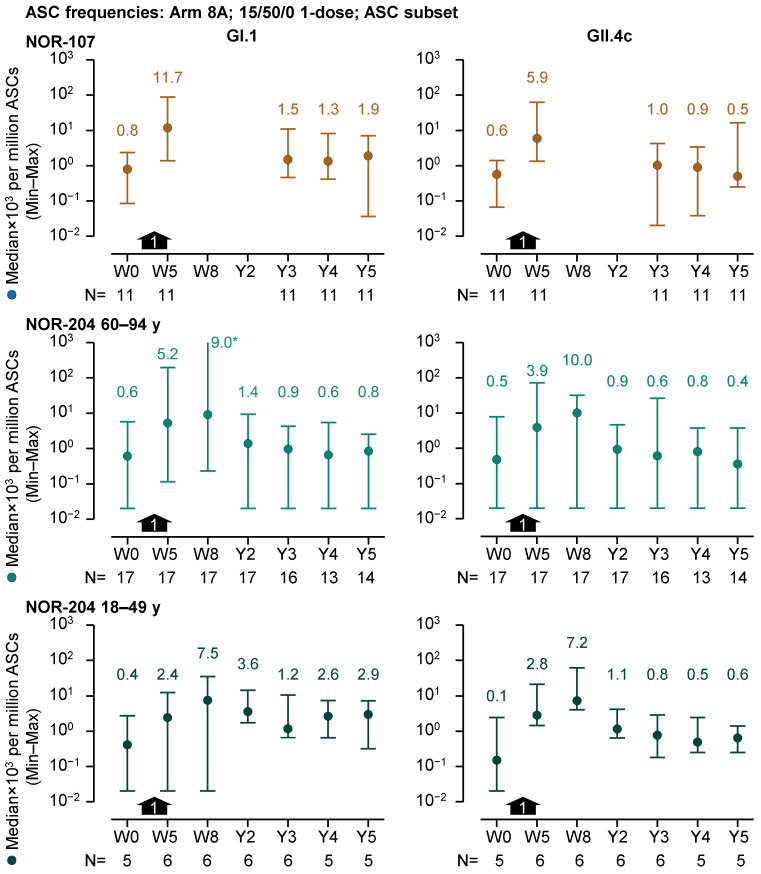
GI.1-specific and GII.4c-specific median antibody-secreting cell (ASC) frequencies per million antigen-secreting cells (ASCs; left and right graphs, respectively) over a five-year period from prior to dosing (Week 0, W0) in the preceding trials to the end of the present follow-up study (Year 5, Y5) for recipients of a single dose of 15/50/0 (Arm 8A) at W4 (black arrows) in the ASC subset. Medians and minimum and maximum values are represented by colored round symbols (and value above the plot) and error bars, respectively. * The maximum value reported at this time point was 2.3 × 10^6^ and hence potentially erroneous (>million GI.1-specifc ASCs per million ASCs). Note, the median ASC frequencies in the figure are expressed in a scientific notation that includes a 10^3^ multiplier. Note also that blank entries in the graphs reflect that sampling was not performed at those time points (i.e., in NOR-107).

**Table 1 vaccines-13-00082-t001:** Arm assignment by former trial and HIL-214 composition and doses.

Arm	HIL-214 Components (µg)				Doses	Number of Participants from Former Trial ^b^			Total Number of Participants
	GI.1	GII.4c	Al(OH)_3_	MPL		NOR-107	NOR-210	NOR-204	
1A	15	15	500	50	1	25			25
2A	15	50	500	50	1	19			19
3A	50	50	500	50	1	27			27
4A	15	15	500	15	1	27			27
**5A ^a^**	**15**	**50**	**500**	**15**	**1**	**23**		**29**	**52**
6A	50	50	500	15	1	27			27
7A	15	15	500	0	1	25			25
**8A ^a^**	**15**	**50**	**500**	**0**	**1**	**28**	**24**	**53**	**105**
9A	50	50	500	0	1	22			22
10A	50	150	500	0	1	28			28
11A	15	50	167	0	1	21			21
5B	15	50	500	15	2			35	35
**8B ^a^**	**15**	**50**	**500**	**0**	**2**	**25**		**42**	**67**
10B	50	150	500	0	2	24			24
11B	15	50	167	0	2	24			24

^a^ Shaded rows with bold text highlight the arms that were the focus of the immunogenicity analyses because these arms evaluated, in more than one former trial, the VLP composition (15 µg GI.1 and 50 µg GII.4c) selected for further clinical development. ^b^ NOR-107 was conducted at two sites in Belgium, NOR-210 was conducted at one site in the US, and NOR-204 was conducted at multiple sites in the US.

**Table 2 vaccines-13-00082-t002:** Demographic characteristics in Arms 5A, 8B, and 8A of the PPS.

Category		5A: 15/50/15, 1-Dose	8B: 15/50/0, 2-Dose	8A: 15/50/0, 1-Dose
		N = 52	N = 67	N = 105
		**Mean (SD)**		
Period from 1st Dose to Last Visit (years)		4.9 (0.5)	4.9 (0.4)	4.9 (0.4)
**Age (years)**	Overall	62.3 (16.8)	63.6 (19.4)	53.9 (21.2)
			**n (%)**	
**Age range:**	18–64	30 (57.7)	28 (41.8)	72 (68.6)
**(years)**	65–84	17 (32.7)	29 (43.3)	22 (21.0)
	≥85	5 (9.6)	10 (14.9)	11 (10.5)
**Female**		34 (65.4)	38 (56.7)	50 (47.6)
**Ethnicity:**	Not Reported	23 (44.2)	25 (37.3)	28 (26.7)
	Not Hispanic or Latino	28 (53.8)	39 (58.2)	69 (65.7)
	Hispanic or Latino	1 (1.9)	3 (4.5)	8 (7.6)
**Race**	White	51 (98.1)	64 (95.5)	94 (89.5)
	Black or African American	1 (1.9)	3 (4.5)	8 (7.6)
	Asian	0 (-)	0 (-)	1 (1.0)
	Native Hawaiian or Other Pacific Islander	0 (-)	0 (-)	1 (1.0)
	Multiracial ^a^	0 (-)	0 (-)	1 (1.0)

Demographic characteristics are taken from when the participant signed the informed consent form (ICF) in the primary trial. Percentage of participants was calculated as 100 × n/N, where n = number of participants with the characteristic, and N = total number of participants in the arm. Note, ethnicity was not collected for participants from NOR-107, and Arms 5A and 8B contain participants pooled across trials NOR-107 and NOR-204, whereas Arm 8A contains participants pooled across trials NOR-107, NOR-210, and NOR-204. PPS = per-protocol set. ^a^ Participant checked more than one race option on the case report form.

**Table 3 vaccines-13-00082-t003:** Adverse events during Year 2 to Year 5 after primary vaccination.

	FAS (N = 528)	
	Events, n_e_	Participants, n (%)
Any Adverse Event	26	18 (3.4)
Related to Trial Procedure	1	1 (0.2)
Related to Prior Trial Vaccine	0	0 (-)
Mild	1	1 (0.2)
Moderate	6	3 (0.6)
Severe	19	14 (2.7)
Leading to Study Withdrawal	1	1 (0.2)
Potential Immune-Mediated Event	6	5 (0.9)
Related to Prior Trial Vaccine	0	0 (-)
Leading to Study Withdrawal	0	0 (-)
Serious Adverse Events (SAEs)	21	15 (2.8)
Related to Prior Trial Vaccine	0	0 (-)
Leading to Study Withdrawal	1	1 (0.2)
Deaths (All Unrelated to Prior Trial Vaccines)	6	6 (1.1)

Percentage of participants was calculated as 100 × n/N, where n = number of participants reporting the adverse event, n_e_ = number of events, and N = number of participants. The Year 2 follow-up visit was not applicable to participants previously enrolled in NOR-107. Although the six participants with fatal SAEs did not complete the trial, only one participant (with metastatic renal cell carcinoma) was reported as withdrawn from the trial due to the onset of the SAE. FAS = full analysis set. SAE = serious adverse event.

## Data Availability

The clinical study report (NOR-213) and associated data on which this manuscript is based are available upon request from Takeda Pharmaceuticals International AG, Switzerland.

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
