# Peer review of "Persistence of the Immune Response to an Intramuscular Bivalent (GI.1/GII.4) Norovirus Vaccine in Adults"

_vaccines, 2025, doi:10.3390/vaccines13010082_

Round 1
Reviewer 1 Report
Comments and Suggestions for Authors
1. Leroux-Roels and collogues have made a fascinating piece of research to complete their previous profile on (GI.1/GII.4) norovirus vaccine. , they do not clearly show the demographic data of their participants, such as whether they were free from autoimmune or not; Free from coagulation disorder or not; any have pregnancy or not, and so on as the gobbler faces a big problem with human trials on COVID-19 vaccines which did not include participant from those categories.
2. The durability reaches up to 5 years which conveys very good immunogenicity (at least on the humoral level), but we need to know the threshold of this vaccine/regimen (GI.1/GII.4) which can offer a protective level against norovirus. The manuscript never indicated this crucial point.
3. The manuscript should indicate to which IgG persists up to 5 years and the relationship to the GI.1 or GII4c?
4. Are they asking the participants if (any) have taken norovirus infection during 5 years?
Reviewer 2 Report
Comments and Suggestions for Authors
Dear authors,
Your manuscript “Persistence of the immune response to an intramuscular bivalent (GI.1/GII.4) norovirus vaccine in adults” describes the assessment of the long-term immunogenicity of HIL-214 norovirus vaccine in adult participants up to five years after vaccination.
The manuscript is well written, contains all necessary parts, the conclusions are supported by data, and all limitations of the study are discussed.
However, some issues should be solved:
L39 – who is vaccine’s manufacturer? Should be described here.
L63 – “at least two of the previous trials.” – the reference to these trials should be added here.
Section 3.1 – Why Arms (Groups?) were names as 5A, 8B and 8A? Has this any meaning?
Table 1 – I am sorry but until this part of the manuscript it is still unclear why have you chosen only there (5A, 8A and 8B) arms for the immunogenicity analysis?
L193 – “Median GI.1-specific” – correct, please, to “median GI.1-specific”.
Discussion section – it seems that discussion of possible adverse events (as you mentioned them) should be made here.
Reviewer 3 Report
Comments and Suggestions for Authors
The manuscript by Leroux-Roels, G. et al. is well-written, the results are clear, and the data interpretation and discussion are solid. Their article shows that both VLPs-based vaccines generate a persistent immune response. The data is solid. However, this study (and related clinical trials) has the draw-back that there is no placebo group.
Minor comments:
1. Please have the same scale on the y-axis for all graphs (0-20,000).
2. The text (and corresponding figures) in sections 3.2, 3.3, and 3.4 is unclear about the following question: I am right to interpret that there are no statistical differences between the GMT at W0 and W8? If so, please clarify this in the text. (I think there is a hint about it in the text, but it would be better to be clear about it.
3. There are lots of acronyms that are not explained. Although some of them are obvious, please explain them.
4. Why are W8 and Y2 missing in GI.1 and GII.4c in Figure 4?
5. It is known in the Norovirus field that culturing human norovirus is extremely challenging and, therefore, almost impossible to perform neutralization assays. Hence, it will be illustrative for readers who are not experts in noroviruses to explain why there no neutralization assays were carried out.
6. Please explain if the deaths in Table 3 are related to the vaccination. This point might be obvious to the authors, but if a reader who is not an expert sees Table 3, they might get alarmed.
7. As the authors do recognize that the major flaw of their study is the lack of a placebo control group. Nonetheless, is there a way to compare data from other studies to infer how effective this vaccine is in generating Norovorvirus-specific antibodies?
Reviewer 4 Report
Comments and Suggestions for Authors
The authors report the findings from clinical trials involving a novel norovirus vaccine. This report focuses on vaccine recipients that received a bivalent vaccine consisting of norovirus VLPs GI.1 and GII.4 and assesses long-term immune responses to norovirus antigens. The authors find that HBGA-blocking titers as well as poly Ig titers are maintained and detectable for both VLPs up to 5 years after vaccine administration. Although less significant and underpowered the authors report and exploratory finding that demonstrates in antibody secreting cells from the NOR-107 and NOR-204 studies.
Although a limited study, the findings would be of interest to those interested in determining long-term responses to VLP vaccines. I have very few concerns with the study as written. Below are a few suggestions.
1. Although the authors state that there are limitations to this study including the lack of a placebo control group, is there an opportunity to cite previous findings on natural infections and the antibody titers induced. Or could the hypothesized infected individuals in this study be shown in a separate figure or graph?
2. The authors do a nice job of demonstrating that an immune response to norovirus after vaccination can be maintained for up to 5 years. However, the authors never speculate on the effectiveness of the vaccine and its potential for inhibiting acute gastroenteritis or other health burdens associated with infection or whether antibody titers found at 5 years post-vaccine can neutralize norovirus infectivity. This seems like an important point that should be addressed.
Round 2
Reviewer 1 Report
Comments and Suggestions for Authors
Thank you,
Round 3
Reviewer 1 Report
Comments and Suggestions for Authors
Thank you for proper responses.